A comprehensive review and call for studies on firefly larvae

Riley William B. 1
Rosa Simone Policena 2
Lima da Silveira Luiz Felipe limadasilveiral@wcu.edu 1
1 Department of Biology, Western Carolina University , Cullowhee , NC , United States of America
2 Instituto de Recursos Naturais, Universidade Federal de Itajubá , Itajubá , Minas Gerais , Brazil
Gillespie Joseph
Electronic publication date: 2021 Sep 20
Publication date: 2021
Volume: 9
Electronic Location ID: e12121
Received 2021 Mar 18; Accepted 2021 Aug 16
Copyright: ©2021 Riley et al.
Copyright year: 2021
Copyright holder: Riley et al.
License: This is an open access article distributed under the terms of the Creative Commons Attribution License, which permits unrestricted use, distribution, reproduction and adaptation in any medium and for any purpose provided that it is properly attributed. For attribution, the original author(s), title, publication source (PeerJ) and either DOI or URL of the article must be cited.
License URL: https://creativecommons.org/licenses/by/4.0/

Keywords: Immature stages, Larval biology, Glow-worms

Funding: The Western Carolina University Biology Department Luiz Felipe Lima da Silveira was funded by the Western Carolina University Biology Department. The funders had no role in study design, data collection and analysis, decision to publish, or preparation of the manuscript.

==============================
Background

Fireflies (Coleoptera: Lampyridae) are commonly recognized by adult traits, such as a soft exoskeleton, lanterns and associated glow and flash patterns, but their larval stage is far less appreciated. However, fireflies spend most of their lives as larvae, and adults of most species rely solely on resources previously obtained. Therefore, studying the immature stages is imperative towards a comprehensive understanding of fireflies. This paper reviews and indicates key gaps in the biology of firefly larvae based on available literature.

Methodology

We reviewed the literature on firefly larvae to identify key issues and important taxonomic, geographic, and subject biases and gaps.

Results

We found 376 papers that included information on firefly larvae. Only 139 species in 47 genera across eight of eleven lampyrid subfamilies have been studied during larval stages. These numbers reveal a staggering gap, since 94% of species and over half of the genera of fireflies were never studied in a crucial stage of their life cycle. Most studies on firefly larvae focus on two subfamilies (Luciolinae and Lampyrinae) in four zoogeographic regions (Sino-Japanese, Oriental, Nearctic, and Palearctic), whereas the other subfamilies and regions remain largely unstudied. These studies mainly dealt with morphology and behavior, other subjects remaining greatly understudied by comparison, including habitats, life cycle, physiology and interactions.

Conclusions

Together, these literature biases and gaps highlight how little is known about firefly larvae, and warmly invite basic and applied research, in the field and in the lab, to overcome these limitations and improve our understanding of firefly biology to better preserve them.

Introduction

The charismatic fireflies (Coleoptera: Lampyridae) are famous for their flashing adults, but the nearly 2200 described species worldwide show wide-ranging phenotypic and ecological diversity, including the less appreciated diurnal and dark species (Branham, 2010; Martin et al., 2019). For example, adults may use diverse mating signals, including light—as continuous glows or flashes—and pheromones, alone or in combination (De Cock & Matthysen, 2005; Lewis & Cratsley, 2008; Stanger-Hall et al., 2018). Because signal diversity is associated with sensor morphology, the use of light and pheromones is accompanied by enlarged eyes and complex antennal morphology, respectively (Stanger-Hall et al., 2018). In addition, most firefly species have been regarded as habitat and/or season specialists as adults (Silveira et al., 2020). Although these data bring about interesting questions, the adult stage is but a flicker in a firefly life cycle, and a staggering lack of information on immature stages significantly obscures our knowledge on the biology of fireflies.

Firefly species are usually semelparous, i.e., they spend most of their lives as larvae. The adult stages only last up to a couple of weeks, in which they reproduce and disperse (Faust, 2017). In fact, adults of most species do not eat, aside from occasional nectaring (e.g., Faust & Faust, 2014), so they must consume all they need to survive during the larval stage (McLean, Buck & Hanson, 1972). Except for predatory fireflies (Lampyridae: Photurinae), who lure other fireflies to ambush and prey upon them (De Cock, Faust & Lewis, 2014), adults of most species live off the energetic supply gathered during their larval stage. Since immature stages bear the bulk of a firefly life cycle, it is crucial to study these semaphoronts to pursue a more comprehensive understanding of firefly biology. However, little to no information can be found in the literature on larval stages of most species as compared to adults.

Here, we review the literature on firefly larvae, emphasizing the knowns and unknowns in available data to provide a starting point for beginners in this field. Specifically, our review provides overall information on crucial biological aspects of firefly larvae, and highlights fundamental open questions.

Survey Methodology

Search criteria and selection of key words

To comprehensively review the literature on firefly larvae, we conducted a systematic survey in the following databases: Zoological Records, Scopus, Biological Abstracts, Biological & Agricultural Index Plus, Web of Science, Google Scholar, and Biodiversity Heritage Library. We searched for the following combination of terms: Lampyridae + larva, Lampyrid* + larv*, firefl* + larv*, firefl* + gastropod*, lampyrid* + gastropod*, and firefl*+snail*. We also used keywords in some languages other than English (Glühwürmchen, Glühwürm, Leuchtkäfer, liciérnag*, pirilamp*, vagalum*, luciol*, lucciol*). As such, our search covered mainly the literature in English, Spanish, Portuguese, Italian, and French. Additionally, we tried to access information in studies in other languages, including Chinese and Japanese, to the best of our knowledge, sometimes not from the original source. We did not set a starting date to include even the oldest works. Finally, we checked the scientific literature list at the Firefly International Network (available at https://fireflyersinternational.net/scientific-literature). Most of the information was obtained directly from the main text, sometimes from abstracts. In some cases, however, where we were unable to access even the abstract, the information was obtained from the sources that cited them (i.e., “apud”; see Supplemental File 2). We disregarded papers that merely mentioned firefly larvae without providing biologically relevant information on this subject, as well as conference presentations, theses and dissertations that were subsequently published, and textbooks without primary data on larva.

Data analysis

We addressed the following four questions: (Q1) How many species in each subfamily have at least one study about their larval stages? (Q2) How often are the subfamilies investigated in the literature (i.e., how many studies deal with each subfamily)? (Q3) What is the zoogeographic distribution of the species surveyed in the studies on larvae? (Q4) How often are our predefined subjects addressed in the literature?

To answer these questions, we scored for each of the 376 papers compiled: the species addressed (Q1), and their respective subfamilies (Q2); the zoogeographic location of the species studied (Q3); and the subjects covered in the study (Q4). A given paper could have multiple tags (e.g., more than one zoogeographic region or subject).

We used the most recent valid name for each species. For Q1, only papers with species-level identification were included, while for Q2 any level of identification beyond the family level was accepted. For the subfamily-level assignment, we followed the classification in Martin et al. (2019), with the addition of Chespiritoinae, recently described by Ferreira, Keller & Branham (2020) (total: 11). To address the state of knowledge of species per zoogeographic region, we followed the classification of Holt et al. (2013), as follows: Afrotropical, Australian, Madagascan, Nearctic, Neotropical, Oceanian, Oriental, Palearctic, Panamanian, Saharo-Arabian, Sino-Japanese (total: 11). Upon review, six subjects stood out as being of particular importance to firefly larval biology: morphology—the physical traits of larvae, their function and use in taxonomic and phylogenetic studies; behavior—the general activity of the organisms; habitat—the environments inhabited and what they prefer in terms of factors such as temperature and moisture; life cycle—phenology and development of larvae; interactions—the ecological relationships of firefly larvae with predators, prey and parasites; physiology—the functioning of organs, cells, tissues and bioluminescence of firefly larvae, including biochemistry. The results are summarized in Supplemental File 1 and Supplemental File 2, as well as in barplots (Figs. 3–6) made in R (R Core Team, 2019), using ggplot2 (Wickham, 2009).

Figure 1 Generalized morphology of a firefly larva.

(A) lateral view, insets show key traits. hd: head and cervical membrane; md: mandible; po: pleural organ; pg: pygopodium; ts/tg: tracheal spiracle/tracheal gill; tb: tubercles. (B–E) variation in lantern disposition. The most common pattern is the scheme in (B–C); Pterotus is an example of (D); (E) is only known in Lamprohiza.

Figure 2 Firefly larvae vary from a flat (with tergites laterally expanded) to slender outline, with intermediate forms.

The legs can be short to very long. Aquatic species often have reduced sclerites, sometimes divided into individualized plates, and might have apneustic or metapneustic gills. (A–H) illustrates the diversity of larval outlines: (A) long legs and laterally expanded terga (e.g., Cratomorphus); (B) terga with tubercles (e.g., Luciola hypocrita); (C) terga narrow with lateral pleurae visible in dorsal view (e.g., Photinus); (D) pleurae dorsally visible with paired tracheal gills (e.g., Aquatica); (E–F) elliptical larvae with terga expanded (e.g., Photuris and Lamprigera); (G) arboreal larvae are usually slender and somewhat flat (e.g., Pyrocoelia atripennis); (H) fossorial larvae are somewhat cylindrical (e.g., Photinus).

Results

We compiled 376 papers that addressed the biology of larval fireflies (Fig. 1). We explored below the knowns and unknowns of larval biology, with special regard to taxonomic, geographic and subject biases in the literature. A full list of literature is provided in Supplemental File 1 and Supplemental File 2.

Taxonomic bias

Of the ca. 2200 species of fireflies across 11 subfamilies, 139 species had at least one study with information on larval stages, across 8 of 11 lampyrid subfamilies—roughly 7% of the total species diversity (Fig. 3). These 139 species span 47 genera, slightly less than half of all lampyrid genera. In three firefly subfamilies, larvae are unknown, revealing a staggering knowledge gap on firefly larval biology. Among these 139 species with known larval stages, 60 are luciolines, 52 are lampyrines, and the remaining 27 species are scattered over six subfamilies: Photurinae (10), Lamprohizinae (6), Ototretinae (3), Cyphonocerinae (1), Psilocladinae (1), Pterotinae (1), in addition to 5 Lampyridae Incertae sedis species (Supplemental File 1 and Supplemental File 2). When the distribution of studies per subfamily is taken into account, a similar bias can be found (Fig. 4). Nearly 80% of the studies concentrate in Luciolinae (192) and Lampyrinae (157), followed by Photurinae (50), whereas the other groups have received less attention: Lamprohizinae (12), Ototretinae (7), Psilocladinae (2), Pterotinae (2), Cyphonocerinae (1), and Incertae sedis (17) (Supplemental File 1 and Supplemental File 2).

Figure 3 80% of species with any study on their immature stages are in two subfamilies: Luciolinae and Lampyrinae, as revealed in a plot of the number of species with at least one study with information on their immature stages.

Figure 4 About 80% of studies on firefly larvae deal with species in two subfamilies, Luciolinae and Lampyrinae, as revealed in a plot of the number of studies per subfamily.

Geographic bias

Most studies on firefly larvae were undertaken in the Sino-Japanese (118), Oriental (88), Nearctic (95), and Palearctic (79) zoogeographic regions (Fig. 5). The other regions—Neotropical (25), Oceanian (12), Australian (11), Panamanian (6), Saharo-Arabian (4), Afrotropical (3), and Madagascan (1)—were far less studied. (Fig. 5; Supplemental File 1). Even though we have not controlled for differences in species richness among regions, regions among the richest in firefly diversity (Neotropical, Panamanian, Afrotropical—cf. McDermott, 1966) are clearly the ones with fewer studies on larval biology, underlining the geographic bias.

Figure 5 Nearly 85% of studies on firefly larvae are concentrated in four zoogeographic regions: Sino-Japanese, Nearctic, Oriental and Palearctic –as revealed in a plot of number of studies per region.

Subject bias

There was a clear bias towards larval morphology (177), followed by behavior (115), while information on habitat (70), life cycle (81), physiology (68), and interactions (39) were less studied (Fig. 6; Supplemental File 1).

Figure 6 Morphology and behavior are the most commonly addressed aspect of firefly biology (53% of the studies), as revealed in a plot of number of studies per subject.

Discussion

Morphology

Firefly larvae have a unique combination of characters among beetles (Figs. 1–2). These larvae tend to have hardened sclerites—regionalized plates of the exoskeleton—on the dorsum (i.e., terga), venter (i.e., sterna) and sides (i.e., pleurites), which contrast with the more lightly sclerotized body of the adults. However, at least some genera have less sclerotized and/or narrower sclerites (e.g., Stenocladius Fairmaire; Ohba, Goto & Kawashima, 1996). Other typical noteworthy traits of firefly larvae include: (i) a retractable head within a prothoracic vault, (ii) sulcated mandibles connected to venom glands, (iii) defensive eversible organs along their sides, (iv) at least a pair of lanterns on their abdomen, and (v) pygopodia—a bundle of membranous tubules outlined by rows of microscopic hooks—on their hindmost segment (Fig. 1A). Nevertheless, not all firefly larvae have all these traits, and some of these can be found in other beetle families as well. For instance, bioluminescence occurs in other beetle families (e.g., Phengodidae, Rhagophthalmidae, some lineages of the Elateridae) (Bocakova et al., 2007; Fallon et al., 2018; Bi et al., 2019). Yet, it is our understanding that the tubular pygopodia with microscopic hooks has only been reported in lampyrid fireflies. Lawrence et al. (2011) coded this trait as “multitubular holdfast organs”, and mention that they are known in all lampyrid larvae, but noted that it has evolved independently in a few staphylinid genera.

The head is connected to the prothorax by a membranous neck, an extensible tubular membrane—often bearing elongate sclerites –, which forms a double layered sheath inside the prothorax vault when the head is retracted (Fig. 1A). When the neck is extended it allows the head to be protruded far ahead of the prothorax, a state seen when larvae eat shelled gastropods (Madruga-Rios & Hernández-Quinta, 2010; Faust, 2017; Vaz et al., 2021), introducing their heads well within the gastropod shell. Larvae can quickly retract their heads in a defensive response and keep them retracted when resting (Fig. 2). The head and its appendages are highly variable across species, particularly the antennae and mouthparts. All known larvae have channeled mandibles connected to venom glands (e.g., Costa, Vanin & Casari-Chen, 1988; Rosa, 2007; Fu, Ballantyne & Lambkin, 2012a; Fu, Ballantyne & Lambkin, 2012b; Vaz, Silveira & Rosa, 2020). The mandibles might be crossed or largely overlapping in frontal view, and often bear one or two teeth. Nevertheless, the biological meaning of all these mandibular variations in head morphology remains unknown.

Some species (including several Luciolinae and Lampyrinae) have pleural defensive glands along lateral regions of thorax and abdomen, dorsal to tracheal openings, which can be everted as a membranous, single or forked tube that is usually stellate at the tip (Fig. 1A). When disturbed, those glands are everted and release a strong smell as well as a repellent substance (Fu et al., 2007; Fu et al., 2009), but sometimes only specific threats will trigger the eversion of these glands (e.g., ants—Tyler, 2001a; Tyler, 2001b; Tyler & Trice, 2001). Pleural defensive glands have been discovered fairly recently, studied in just a handful of species, and their biological meaning for firefly ecology and evolution is yet to be understood.

The pygopodia (Fig. 1A) can be used as hooks to attach to substrates and aid in movement on land and grooming (Archangelsky & Branham, 1998), as well as to adhere to floating substrates in water (Fu et al., 2005a). Variation in pygopodial morphology has been observed in the number of basal stalks and hook arrangements (e.g., Fu, Ballantyne & Lambkin, 2012b), but the meaning of such variation still needs to be addressed.

All known firefly larvae are bioluminescent, with the possible exception of an undetermined species—lanternless and allegedly dark—recently reported from the Amazonian tepuis by Kok, Van Doorn & Dezfoulian (2019). Bioluminescence is produced in the paired lanterns (also known as photogenic organs), usually located on the abdomen (Fig. 1). Most species have only one pair of lanterns as spots on the ventral surface of the abdominal segment VIII (Figs. 1A, 1B), while some others have paired spots on VII and VIII (Fig. 1D, e.g., Pterotus LeConte, 1859; Stenocladius Faimaire 1878), on the pronotum (Nunes et al., 2021), or feature polymorphic patterns of single or paired spots on segments II-VI (Fig. 1E, Lamprohiza Motschulsky 1853; Novák, 2018a). The light becomes more visible when the larva twists or arches their posterior end, but some species have translucent areas in the tergites above the lanterns that make the light visible above the body (Fig. 1C, e.g., Psilocladus Blanchard, 1846, Lamprigera Motschulsky, 1853) (e.g., Costa, Vanin & Casari-Chen, 1988; Novák, 2018a; Novák, 2018b; Vaz, Silveira & Rosa, 2020).

When comparing the general larval morphology across firefly species, a few patterns arise, which are associated with their habitats and lifestyles. For example, fossorial larvae tend to be rounded, with more extensive membranous areas (Figs. 1A, 2C, 2H, e.g., Photinus Laporte, 1833; Phosphaenus Laporte, 1833), while leaf litter dwellers are flatter, with laterally expanded terga (Figs. 2A, 2B, 2E, 2F; e.g., Photuris LeConte, 1852; Cratomorphus Motschulsky, 1853; Lamprohiza). Some tree-climbing species are very narrow and flat, with elongate legs, reduced pleural regions, and ventral surface well-sclerotized (Fig. 2G; e.g., Pyrocoelia atripennis Lewis, 1896). It is therefore tempting to think that their overall twig-like morphology is involved in camouflage—a hypothesis yet to be tested.

Aquatic species tend to resemble fossorial larvae in outline (i.e., cylindrical, with narrow terga, exposed pleura) (Jeng, Lai & Yang, 2003; Fu, Ballantyne & Lambkin, 2012b). Most terrestrial (e.g., Photuris spp. –Rosa, 2007) and semiaquatic (Aspisoma fenestrata Blanchard, 1839; Archangelsky, 2004) firefly larvae have typical insect spiracles as respiratory openings, but aquatic larvae are adapted to exchange gases under water by tracheal gills, apneustic (i.e., without spiracles) or metapneustic (with spiracles on the terminal gill) (Fig. 1A, e.g Aquatica Fu, Ballantyne & Lambkin, 2010, and some Luciola Laporte, 1833) (e.g., Jeng, Lai & Yang, 2003; Fu, Ballantyne & Lambkin, 2012b; Ballantyne & Lambkin, 2013). Aquatic backswimmers (e.g., Sclerotia substriata (Gorham, 1880) and S. aquatilis (Thancharoen et al., 2007)) lack gills like the semiaquatic larvae, which have distinctly harder exoskeletons, and use their pygopodia to adhere to floating objects (Fu, Wang & Lei, 2005; Fu et al., 2005a; Fu et al., 2005b; Fu et al., 2007; Fu et al., 2009; Fu, Ballantyne & Lambkin, 2012b; Sato, 2019). Interesting ontogenetic shape changes have been reported for the lucioline genus Sclerotia Ballantyne, 2016, where apneustic early larvae become metapneustic in later instars (Fu, Ballantyne & Lambkin, 2012b). That shape change is followed by a change in mode of life, since benthic apneustic larvae become backswimmers.

Some truly aquatic larvae might also feature less sclerotized sclerites, sometimes with terga completely split into two rounded plates (Fig. 2D, e.g Aquatica –cf. Jeng, Lai & Yang, 2003; Fu, Ballantyne & Lambkin, 2012b; Micronaspis Green, 1948 –Vaz et al., 2021). Highly sclerotizated exoskeletons in arthropods are associated with increased resistance to water loss, but might also hamper cutaneous respiration. Moreover, these are costly to produce, especially where calcium ions are harder to obtain, such as in freshwater environments (Brusca, Moore & Shuster, 2016). Therefore, at least in aquatic lucioline larvae, a less sclerotized body might be the outcome of adaptation to life in freshwater. Interestingly, a fully aquatic lifestyle evolved multiple times in Lampyridae, and possibly at least twice within the Luciolinae (Jeng, Lai & Yang, 2003; Fu, Ballantyne & Lambkin, 2012b).

Larvae of some distantly related genera (e.g., the luciolines Abscondita Ballantyne, Lambkin & Fu, 2013 and Pygoluciola Wittmer 1939, and the lampyrine Micronaspis) bear tergal and pleurite tubercles of unknown function (Figs. 1A, 2B). It has been proposed that such tubercles might help Micronaspis attach to the substrate—thus avoid being dragged by sea currents –, fend off predators, or both (Vaz et al., 2021; Nada, Ballantyne & Jusoh, 2021).

Sexual dimorphism in larvae was only reported for a single species, Luciola discicollis Laporte, 1833, from Ghana (Kaufmann, 1965). Interestingly, that species features sexually dimorphic larval lanterns that are larger in males. This trait has been overlooked in other species, since one would need to separately raise and keep exuviae or detailed images to compare after identifying the sex of the emerged adults.

Species in most genera across subfamilies cannot be reliably identified based on larval stages, mostly due to a staggering lack of comparative morphological studies. However, it is noteworthy that larval traits have been successfully used not only to discriminate between species, but as relevant sources of phylogenetically informative traits (Ballantyne et al., 2019; Vaz et al., 2021).

Understanding the evolution of firefly larval traits has been precluded by two main factors: a lack of descriptive and functional studies, and unsteady phylogenetic frameworks within the Lampyridae, and of the Elateroidea as a whole. The limited breadth of lampyrids ever studied in the larval stage (∼7% total species), on top of the staggering taxon-bias (see above), limits any generalisation regarding trait evolution.

Above all, phylogenetic analyses of the Lampyridae have had a relatively limited taxon sampling, not necessarily covering the taxa with known larval stages, and the resulting proposals of classification have been unsteady (Branham & Wenzel, 2003; Martin et al., 2019). To compound the issue, the relationships within Elateroidea have been ever changing, often in conflict with traditional classification schemes. For example, earlier comprehensive DNA-based phylogenies on Elateroidea found Lampyridae to be sister to either Cantharidae (Bocakova et al., 2007; Kundrata, Bocakova & Bocak, 2014; Bocak et al., 2018) or to Phengodidae + Rhagophthalmidae (i.e., the “Lampyroidea” hypothesis; McKenna et al., 2015), whilst more recent and advanced approaches have confirmed Lampyridae as sister to the clade Phengodidae + Rhagophthalmidae either sister to or within Elateridae (McKenna et al., 2019; Kusy et al., 2021; Douglas et al., 2021). Together, these limitations hamper any detailed, comprehensive evolutionary analysis of firefly larval traits.

A few interesting patterns emerge when considering “Lampyroidea” within Elateridae, and the Lampyridae as a whole. For instance, the ability to emit light (at least in the larval stages) has been proposed as a putative synapomorphy for Elateridae including the “Lampyroidea” (cf. Douglas et al., 2021). Furthermore, the channeled larval mandibles are a putative synapomorphy of the “Lampyroidea”, even though the condition allegedly evolved independently in Brachypsectridae, Cerophytidae + Jurasaidae, and the elaterid lineage Drilini, as well as and in several adephagan lineages (Beutel, 1995; Costa et al., 2003; Lawrence et al., 2011; Rosa et al., 2020).

Interestingly, firefly larvae of at least a couple of species (e.g., Lampyris noctiluca (Linnaeus, 1758)) show bioluminescence “leakage” (i.e., invisible to human eyes) through paired lateral spots (over which the exoskeleton is translucent) that match those seen in phengodid larvae (Tisi et al., 2014). These spots have been interpreted as casts of ancestral lanterns, and might be another potential synapomorphy of the “Lampyroidea”, although these could also represent an atavic condition. Finally, putative exclusive synapomorphies of Lampyridae include the pygopodia, the pair of lanterns on sternite VIII, and the head fully retractable into the prothorax.

Behavior & Interactions

Given the scarcity of studies detailing interactions among firefly larvae and other organisms, behavior and interactions are grouped together.

Larval bioluminescence has been consistently observed as an aposematic warning signal (Underwood, Tallamy & Pesek, 1997; De Cock & Matthysen, 2001; De Cock & Matthysen, 2003), but other functions have been proposed as well (e.g., illumination or intra-specific communication—Sivinski, 1981; Buschman, 1988). Vertebrate predators learn to avoid firefly larvae by associating their glows to unpalatability (Underwood, Tallamy & Pesek, 1997; De Cock & Matthysen, 2003). Many firefly species have a patchy distribution in the larval stage (e.g., Kaufmann, 1965; Kakehashi, Kuranishi & Kamata, 2013), and seem to agonistically glow in clusters, as if the group was amplifying the visual signal ( L Silveira, 2021, pers. obs.)—a hypothesis yet to be tested.

Firefly larvae are predators with extra-oral digestion, and a notorious preference for soft bodied invertebrates, notably gastropods. In fact, specialization in gastropods is so extreme that firefly larvae can recognize the chemical signature of snail and slug slime to decipher their direction (Fu & Meyer-Rochow, 2013; Sato, 2019). Even marine fireflies prey on gastropods (e.g., Micronaspis gabrielae Vaz, Silveira & Rosa, 2020—Vaz et al., 2021). Firefly larvae mastered gastropod-eating through a menagérie of complex behaviors, including snail-riding (climbing the shell and biting from above, e.g., Lampyris noctiluca—Tyler, 2002), snail-lifting (lifting the snail and holding it in the air before biting, e.g., Micronaspis gabrielae—Vaz et al., 2021), and tracking of mucus trail (Pyrocoelia atripennis—Sato, 2019). At least under lab conditions, firefly larvae of many species would feature “congregating feeding behavior”, upon which several larvae will eat the same prey (e.g., Fu & Meyer-Rochow, 2012; Nunes et al., 2021). This phenomenon was never documented in the field, and is still largely unstudied.

Alternatively, the fossorial larvae of several Photinini LeConte, 1881 species are known to eat (e.g, Hess, 1920; Archangelsky & Branham, 2001) or even specialize in earthworms (e.g., De Cock, 2000), while at least some Photuris species are more generalist, eating gastropods, earthworms, termites and even decaying matter (Buschman, 1984; Rosa, 2007).

Larvae will use the sulcate, distally opened mandibles to pierce and inject a supposedly neurotoxic venom to paralyze their prey (Fig. 1A) (Williams, 1917b; Hess, 1920; Fabre, 1924; Tyler, 2002; Walker et al., 2018). Once the prey is paralyzed, the prey’s flesh will be cut by the larva’s mandibles into smaller pieces, dissolved with digestive enzymes that are regurgitated from the midgut, then sucked into the oral cavity (e.g., Sato, 2019). Since firefly larvae are very effective at eating gastropods, the former have been proposed as biological controllers of the latter (e.g., Fu, Wang & Lei, 2005; Fu & Meyer-Rochow, 2012; Fu & Meyer-Rochow, 2013; Fang et al., 2013; Sato, 2019). Despite the overall enthusiasm with that possibility, studies measuring the success of firefly larvae as biological controllers at agricultural scales are lacking.

The range of activity of adult fireflies is very broad, with some being active only at the dusk or late night, some only for short spans, while others are active for hours on end (Madruga-Rios & Hernández-Quinta, 2010; McLean, Buck & Hanson, 1972). However, larval behavior is far less researched than that of the adults; larvae are assumed to be almost strictly nocturnal but might have been overlooked during the day. In fact, Lampyris larvae have been spotted crawling in daytime, searching for pupation sites (Tyler, 2002). During the day, terrestrial larvae are presumed to rest in soil, leaf litter, and under stones (McLean, Buck & Hanson, 1972; Faust, 2017), rock crevices (Vaz et al., 2021), or inside bromeliads (Vaz, Silveira & Rosa, 2020). At night they appear on the surface, crawling among grass, weeds, and sometimes climbing up a few inches on stems, particularly if the weather is damp (Faust, 2017).

Aquatic firefly larvae dwell in slow-moving rivers and ponds. They either anchor themselves with their pygopodia or swim near the surface in a squirming back swimming motion to search for prey (Jeng, Lai & Yang, 2003; Fu, Ballantyne & Lambkin, 2012b). Larvae of at least some aquatic species (e.g., Aquatica ficta –Ho et al., 2010) float on water and take on currents, which helps larvae disperse within the same habitat. Moreover, the salt-water adapted Micronaspis floridana Green, 1948 larvae were observed taking on a “canoeing” behavior, upon which the larva would better float (Vaz et al., 2021), and possibly disperse on sea currents. Interestingly, aquatic larvae will climb to land and search for pupation sites off water (Ho et al., 2010).

Semi-aquatic larvae dwell in the soil and leaf litter on river banks and pond margins, but move to the water for short periods when foraging (e.g., Fu et al., 2005a; Fu et al., 2006a). Tree-climbing larvae will often dwell on the ground but will climb trees when tracking prey, by following gastropod mucus trails (e.g., Madruga-Rios & Hernández-Quinta, 2010; Sato, 2019). SR and LS have oftentimes found fossorial and tree-climbing firefly larvae in the Atlantic Rainforest (Brazil), but never managed to raise these to the adult stage.

Firefly larvae are chemically defended and aposematic, which usually protects them from generalist predators (De Cock & Matthysen, 2001). In fact, only a few predators of terrestrial firefly larvae have been reported—mostly birds (Lloyd, 1973). In freshwater, scyllarid lobsters, stream crabs, dragonfly larvae, and Taiwanese tilapia are known predators of aquatic firefly larvae (Ho & Chiang, 2002 apud Ho et al., 2010). Predators that specialize in firefly larvae were never reported. However, a wide array of parasites of firefly larvae has been reported (Lloyd, 1973; Day, 2011; Faust, 2017). These include entomopathogenic fungi (cf. Day, 2011; Foo, Seelan & Dawood, 2017), nematodes, and parasitoid tachinid and phorid flies (Lloyd, 1973; Faust, 2017). Interestingly, but consistent with their mostly diurnal behavior, parasitic wasps were never reported to parasitize firefly larvae.

To avoid enemies, firefly larvae often have color patterns that match their backgrounds (cf. Faust, 2017). Some species will go dark and feign death if discovered by a potential predator (e.g., Darwin, 1859; Fu et al., 2007). Otherwise, larvae typically glow when disturbed, as a warning to potential predators, or evert repugnatorial glands if they can (Fu et al., 2009).

Firefly larvae often carry many mites, especially in the prothorax vault (Faust, 2017; L Silveira, 2021, pers. obs.). These mites are likely commensals, living on the leftovers of firefly larvae. Likewise, fireflies can be commensals on nests of termites or ants. Larvae of some North American Pleotomodes Green, 1948 and the North African Pelania mauritanica (Linnaeus, 1767) have been found at the entrance or in chambers within ant nests, but no obvious positive interactions between lampyrids and ants are known (Cros, 1924; Sivinski et al., 1998). These lampyrid larvae do not feed on ants, and the ants do not seem to notice the presence of the larvae or adult fireflies that live in their nests (Cros, 1924; Sivinski et al., 1998). These ant nests might offer damper conditions than those found outside, given that these firefly species occur in arid environments. Undescribed larvae of a Psilocladus sp. (Psilocladinae) and another unidentified species (possibly a Lampyrinae: Cratomorphini), have been found inside termite nests in Brazil (S Rosa, 2021, pers. obs.). These later are entirely unpigmented, a common feature in fossorial beetle larvae (e.g., cantharid Macromalthinus Pic, 1919 (Biffi & Casari, 2017) and Jurasaidae species (Rosa et al., 2020)). Under laboratory conditions, Psilocladus larvae of two species studied by our group readily ate termite workers, consistent with the possibility that their relatives live on termites in the field (Vaz, Silveira & Rosa, 2020; S Rosa, 2021, pers. obs.).

Matriphagy has been hypothesized in at least one species, Phausis inaccensa LeConte, 1878, where larviform females partake in egg guarding behavior (Faust & Forrest, 2017). In that species, larvae refused to eat food items readily eaten by firefly larvae under lab conditions. Given that the dead body of the mother was separated from the offspring, the authors proposed that matriphagy might be important in normal development of P. inaccensa larvae—a hypothesis yet untested.

Habitat

Firefly larvae are adapted to a diverse array of microhabitats. Terrestrial larvae might be tree-climbers, crawl over or under the leaf litter, or even be fully fossorial—for each microhabitat typical morphological traits can be associated (see above). Terrestrial and semi-aquatic larvae will often be found under rotten logs or rocks, but one species—the only described psilocladine larvae—is associated with bromeliads (Vaz, Silveira & Rosa, 2020). Aquatic larvae live in streams, ponds, mangroves, rocky outcrops, and aquatic crop fields; dwell on the bottom, or are backswimmers (e.g., Ho et al., 2010; Vaz et al., 2021). Both terrestrial and aquatic species are known to build hypogeous, mud pupal chambers (e.g., Bicellonycha Motschulsky, 1853—Cicero, 1982; Aquatica—Ho et al., 2010). Many others pupate on leaflitter (e.g., Cratomorphus—Campos, Silveira & Mermudes, 2018), or hang off the ground by the pygopodia on vegetation (e.g., Pyractomena - Archangelsky & Branham, 1998). Interestingly, because most firefly larvae specialize in gastropods, habitat displacement could have contributed to lampyrid radiation, a hypothesis never formally addressed.

The habitat requirements for larval survival are often different from those of adults, which must be taken into account in conservation policies. In other words, the lack of knowledge about larval needs might undermine the success of the conservation firefly species.

Life cycle

Fireflies live for approximately one or two years, depending on the species, usually spending most of that time as larvae. Under laboratory conditions, larvae raised from eggs usually took 2–11 months to reach pupal stage, through 4–7 instars (Bess, 1956; Rosa, 2007; Ho et al., 2010; Viviani, Rosa & Martins, 2012). However they may take even longer to pupate. For example, larvae of the firefly Pyrocoelia pectoralis have rather plastic life cycles that alternate between 1–2 years (Fu & Meyer-Rochow, 2013).

Most species living in temperate climates hatch in the summer, and live as larvae until spring before pupating. Known exceptions are Ellychnia corrusca, which pupate in the fall, as well as Pyractomena borealis and some European Lampyrini, which pupate during spring (Álvarez & De Cock, 2011; Faust, 2012). Seasonality of tropical species is far less understood. A few tropical multivoltine species have been reported, which are known to pupate in the rainy season (Vaz, Silveira & Rosa, 2020; Vaz et al., 2021).

Physiology

Few studies have addressed the physiology of firefly larvae to various degrees. However, these span a wide range of topics, including larval respiration (e.g., Buschman, 1984), molecular properties of bioluminescence (e.g., Viviani et al., 2004; Tisi et al., 2014), neurophysiology (e.g., Vencl et al., 2012), genome-wide adaptations to aquatic lifestyles (Zhang et al., 2019), and the developmental origin of larval lanterns (Hess, 1922; Tonolli et al., 2011; Stansbury & Moczek, 2014). Otherwise, all basic aspects of larval firefly physiology remain unknown (e.g., venom synthesis and metabolism, digestion, hormone signaling and molting, osmoregulation and metabolic waste management, etc.). Only a few species were investigated in studies of firefly larval physiology. Thus, a broader taxonomic sampling is sorely needed to assess the level of generalization of these findings.

Fireflies are known to inject venom and to extra-orally digest prey with saliva, but the exact composition, synthesis, and variations in these fluids remain unknown. Furthermore, freshwater, estuarine and even marine species of fireflies exist, but how they cope with ion regulation is yet to be discovered. Importantly, the physiology of molting in firefly larvae—which would provide insight into the widespread paedomorphosis in fireflies—remains unexplored (but see Chanchay et al., 2019).

Buck (1948) addressed the physiology of firefly bioluminescence and summarized early scientists’ contributions on this topic. Early works on the source of firefly light contrasted two hypotheses: endogenous versus bacterial (Williams, 1917b; Harvey & Hall, 1929; Hasama, 1942). The endogenous hypothesis has gained consensus after the discovery of luciferin, luciferase and their chemical reaction. Tisi et al. (2014) demonstrated the presence of low-level bioluminescence throughout the body in all life stages of L. noctiluca suggesting a secondary role of luciferase.

Carlson (1965), Carlson (1968a), and Carlson (1968b) investigated the bioluminescence of Photuris larvae under effects of drugs, electrical stimulation and oxygen concentration to compare larval and adult pseudoflashes (i.e., artificially simulated). His results were seminal to later research on the neurophysiological control of Photuris lanterns (e.g., Oertel & Case, 1976; Christensen & Carlson, 1981; Christensen & Carlson, 1982; Vencl et al., 2012). Of particular interest was the finding that larval glows were induced by irritation of lateral abdominal setae, and mediated by octopamine, which is supposed to be homologous to vertebrate noradrenaline (Vencl et al., 2012).

Harvey & Hall (1929) have ablated the larval lanterns of an unnamed species, in an experiment suited to test whether the larval light was endogenous or bacterial. The few individuals that survived to the adult stage had wild type lanterns. Assuming that these researchers successfully ablated the larval lanterns, their results suggest that the adult lantern tissues are not developed from larval ones. Thus, adult lanterns might not be homologous from the developmental perspective. Future studies addressing the evolutionary and developmental origin of larval and adult lanterns are of utmost importance to understand the evolution of fireflies. For example, whether the larval lantern tissues of the “lampyroid” families Phengodidae and lampyridae are homologues, to our knowledge, is an open question.

Conclusions

The information available on firefly larvae is rich and valuable, although heavily concentrated on a few taxa, regions, and subjects. Luciolinae is by far the best known subfamily from the perspective of larval biology, and Asia is the region best surveyed. Most studies on firefly larvae deal with morphology, followed by behavior. Other subjects remain greatly understudied by comparison, including habitat, life cycle and interactions. Together, these literature biases and gaps highlight how little is known about firefly larvae.

The huge overarching gaps in firefly biology have many problematic implications. For example, identification of firefly larvae is currently impossible without raising them to adult stage. On one hand, species discrimination among firefly species is mostly dependent upon adult male traits (e.g., McDermott, 1964). On the other hand, identification methods based on DNA (e.g., DNA barcoding) depend on comprehensive taxon and geographic sampling (e.g., Joly et al., 2014), both currently lacking in available databases and literature on fireflies. Therefore, the lack of information on larval morphology and general biology renders species identification of immature stages virtually impossible.

Ultimately, we need more information on firefly larvae to more deeply understand the evolution of the Lampyridae. How do we identify larval fireflies? Where and for how long do they live? When are they active and not? What do they eat? What’s in their venom? How do ontogenetic shape changes correlate with larval ecology? How do all these traits evolve within and across firefly lineages? Moreover, how much of firefly diversity was driven by adaptive processes associated with larval biology? Do larval and adult lanterns have the same developmental origin? Future studies targeting these issues are sorely needed to overcome knowledge shortfalls and pave the way towards a more comprehensive understanding of fireflies.

Supplemental Information

Supplemental Information 1 Synoptic list of references on Lampyridae larva, arranged by subfamily and zoogeographic region, and scored by subject. A full list of References is given below

Abbreviations indicating the main topic of the works are as follows: B, behavior; H, habitat; I, interaction; L, life cycle; M, morphology; P, physiology.

Click here for additional data file.

Supplemental Information 2 Reference list indicating the species included in each study

Click here for additional data file.

We thank Bruna P. Oliveira for drawing Fig. 1A, J Brawley and S Hoskins for critically reviewing the manuscript.

Additional Information and Declarations

Competing Interests

Author Contributions

Data Availability

The authors declare there are no competing interests.

William Riley conceived and designed the experiments, performed the experiments, analyzed the data, authored or reviewed drafts of the paper, and approved the final draft.

Simone Rosa and Luiz Felipe Lima da Silveira conceived and designed the experiments, performed the experiments, analyzed the data, prepared figures and/or tables, authored or reviewed drafts of the paper, and approved the final draft.

The following information was supplied regarding data availability:

This is a literature review.

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
