# Peer review of "A comprehensive review and call for studies on firefly larvae"

_PeerJ, doi:10.7717/peerj.12121_

## Round 0.1 · original submission · Major Revisions

Dear Dr. Riley and colleagues:

Thanks for submitting your manuscript to PeerJ. I have now received three independent reviews of your work, and as you will see, the reviewers raised some concerns about the research. Despite this, these reviewers are optimistic about your work and the potential impact it will lend to research on firefly systematics. Thus, I encourage you to revise your manuscript, accordingly, taking into account all of the concerns raised by the reviewers.

The manuscript appears to lack a complete synthesis of the system, with no comparison with related groups. Your revision should be less superficial, moving beyond previous statements/concepts by other authors. Furthermore, the organization of concepts in the manuscript appears disjointed. The approach to dividing the World into eight sections was not well-received and indeed appears odd.

Please improve the content and clarity of the figures and tables (see suggestions by the reviewers). Please also ensure that all appropriate references are included.

The majority of these criticisms is highly constructive and should immensely improve your manuscript. Please also note that reviewers 1 and 2 have included marked-up versions of your manuscript.

I look forward to seeing your revision, and thanks again for submitting your work to PeerJ.

Good luck with your revision,

-joe

Reviewer 1 ·

Basic reporting

The authors submitted a review on lampyrid larvae and I would expect that a review means to set a topic into a wide perspective which was not clearly present in original studies dealing with specific questions. The authors did not try to do anything like that.
I expected an approach based on relationships, but the review is a repetition of isolated findings.
Any comparison with other groups, i.e., a wider perspective, is missing.

Concerning PeerJ requirements the manuscript fulfils the criterion for acceptance.
It contributes to the volume of scholarly literature, although it does not contribute to knowledge.

Experimental design

Simple design - the list of literature with findings rewritten.

Validity of the findings

Some 'unpublished' info reported without any hard information and experiment design

Additional comments

I made numerous comments into the manuscript. Hopefully, you find them useful.

Annotated reviews are not available for download in order to protect the identity of reviewers who chose to remain anonymous.

Reviewer 2 ·

Basic reporting

This manuscript introduces an interesting topic about the larvae of fireflies (Coleoptera: Lampyridae). It is a literature review covering 231 publications. Authors identified main important topics related to the firefly larvae and tried to find out the geographical and topical biases and main gaps in the firefly larval research.

Generally, I like the manuscript and topic it presents very much. I think this has a great potential to be a landmark study which can be highly cited by all subsequent researchers on firefly ecology, morphology, biology, behavior, and other aspects. It fits to the scope of PeerJ, too.

The text is generally rather clearly written but I think some additional proof reading by English native speaker would be beneficial (a few sentences hard to read, plural vs singular, etc).

Introduction provides the most important background of the field, however, I think that the last part with results and call for future work does not belong there, as it is more appropriate for Discussion or Conclusions. After removing that last part, Introduction would be quite short, and could be definitely improved by adding information about the geographic distribution, classification, species richness, etc. of Lampyridae to provide the readers a bigger picture.

I do not know exactly the policy of PeerJ about using taxonomic names in manuscripts, but usually author plus year or at least author should be mentioned for each species and genus-name when it appears first time in the text. I keep this on editor whether or not this is needed.

Regarding figures, I really do like figures 1 and 2, which show the morphology of firefly larvae and are really well-drawn. This may attract attention definitely. However, the legends could be improved in my opinion. For example, in Figure 1B-E, some examples of genera or species could be added in order reader better understands where to find these variations in lantern disposition. The same for all drawings in Figure 2. What is more, the legend of figure 2 has some introductory story which should probably go into main text. This might be, however, some format requirement of PeerJ (?) because I see similar explanatory sentences also in Figures 3-5.
Results given in Figures 3-5 can be more user-friendly when they are put in the Table. Alternatively, it would be also interesting to merge these three aspects together, in order readers can see how many papers for example focused on the Luciolinae (aspect 1) behavior (aspect 2) in Asia (aspect 3), or Photurinae life cycle in North America, etc. This would allow better discussion and better presentation of results, and conclusions.

Regarding the organization of main text, there is twice a conclusions part called "Conclusions" (in between Methods and Discussion, which is weird to me), and "Concluding remarks" (at place where it is supposed to be, i.e. at the end of the manuscript, after discussion). These should be probably merged into one.

There are some minor issues in References; they should be double-checked for the required format of PeerJ.

Experimental design

This is to me probably the most problematic part of the presented study. Authors divided World into eight regions and tried to show which regions are understudied regarding the larval fireflies. However, they did not use any of the official realms/regions classifications defined by Wallace and used for ages (i.e, Oriental, Neotropical, Palaearctic, etc.) or still quite recently by Holt et al. (2013). Instead, they divided whole World into eight regions, from which four (half!) is America (South America, North America, Central America, and the Caribbean). On the other hand, for example, whole Asia (i.e., from Turkey and Saudi Arabia through Mongolia to Indonesia, covering many completely different areas in terms of geography, aridity, temperature, flora, etc.) is one region only!
Authors should definitely either provide very good explanation for their division of the World, or change this completely and hence also results, conclusions and respective figure. It seems to me really inappropriate to compare the Caribbean with whole Asia...

Further, I miss for example Liew and Schilthuizen (2014) PeerJ in this review, and there are also some citations therein (which I have not examined, however).

Validity of the findings

There are quite a number of "hypotheses yet to be tested" mentioned in the discussion, these can be probably summarized in the Conclusions part at the end of the main text.

Also, see my remarks on the problematic division of the World into eight proportionally totally different regions by authors, which affected the final conclusions.

Additional comments

Some changes are required in order the manuscript is improved to the higher quality, and conclusions can be better supported by data (i.e. geographic aspect problem).
However, I want to make clear that I definitely support publishing this manuscript (but improved version) in PeerJ!

Major issues are reported in previous sections, and most of my comments/suggestions are embedded directly into the PDF attached.

Annotated reviews are not available for download in order to protect the identity of reviewers who chose to remain anonymous.

Reviewer 3 ·

Basic reporting

Overall, the review manuscript uses clear and professional English throughout, although some contexts can be improved. It contains references to the most up-to-date work on firefly larva but may not be comprehensive enough. It provides sufficient background which fits into the broader field of knowledge about firefly larva's biology aspects with appropriate references.

However, the review manuscript structure does not fully conform to an acceptable format of PeerJ's 'standard sections'. See line numbers 101-105. The Conclusion heading should be after Survey Methodology and before Acknowledgement sections. In addition to that, there is another conclusion section (under subheading). See line numbers 388-408. The headings/sections should be the topic of interest, not the "Discussion".

Figures are relevant to the article's content but not appropriately labelled, particularly category title of axis X and Y of Figure 3-5.

This review is essential and timely because studies on firefly larva are scarce compared to studies published on adult fireflies. The authors made a good introduction to fill the gaps in larval biology knowledge based on published literature.

Experimental design

The Survey Methodology needs a bit more clarity, and from the current form of writing, the sources of literature are not yet exhaustive. A comprehensive review should also include extensive searching, which means that the authors should demonstrate that their reviews are not biased or skewed toward specific browsers or databases when possible. Instead of browsers, one would prefer to start first with the most comprehensive and well-organised databases such as Scopus and Web of Science (although pay-walled). There would be discrepancies, but there would also overlapping results among these databases. The cutoff dates of review (year of publications) are not stated.

The choice of string search or keywords is essential, and usually, we would make it more variable not to be too restrictive. For instance, using Web of Science, "Lampyridae" AND "larva" will return 97 results, but if you use "firefl AND larva" (using asterisk to denote "fireflies or firefly", "larva or larvae"), you will get even more papers. In another instance, using the exact string searches, Scopus returned 109 and 208 results, respectively. If not the entire WOS, the authors might want to look into the Zoological Records database (partly owned by WOS). Also, authors might want to dig deeper into classic studies in Biodiversity Heritage Library (BHL) regarding taxonomy and behaviour studies and not just rely on FIN's website, which is merely a list of references. There are cases where literature is inaccessible because of a pay-wall. It is worth mentioning if some literature came from author's personal copies. From the accumulation of literature, perhaps it would be great to map the critical points in the study of Lampyridae by chronological order: e.g. when the study on larva sparked the interest, and then pinpoint where the gaps start appearing.

The review needs more improvement in its organisation. Please check again the formatting and see examples from other literature review published in PeerJ.

Validity of the findings

This manuscript aimed to review and indicate key gaps in the biology of firefly larvae based on available literature. The authors reviewed 231 papers and categorised the findings based on geographic region, subfamilies and five major themes. Although they have demonstrated an excellent reference to literature, especially on the morphology section, the argument they presented is not well-developed to meet the goal set out in the Introduction. In line 87, "only papers.." it is unclear whether the authors referred to the 231 papers or were the papers examined a subset of 231 papers?

I wasn't sure whether the review is about trends in research of firefly larva or is it about gaps of knowledge in larval biology? It's somewhat in between.

The Conclusions are too vague. As they call for more studies, they did not explicitly outline what gaps we can learn from this review and identify what future directions we could take. The questions in the Conclusions are too general, and the authors already pointed some of them out in the texts. For instance, on what firefly larvae eat, authors have already addressed a few studies that showed evidence of diet type of larvae. However, what is lacking (gap) is that firefly larva is a diet specialist; hence different firefly species would require different prey species. Prey species availability would also depend on the habitat where firefly species thrive. The future studies are challenging and probably still skewed to the well-studied region or subfamilies (which is common)- this is where the authors would provide where best to tackle this issue.

Additional comments

I commend the authors for their excellent effort to come up with a review on firefly larva. Although the manuscript can be considered as a topic of high interest, the current form of writing is not yet ready for publication and it needs major improvement before it can be accepted for publication.

---

## Round 0.2 · Minor Revisions

Dear Dr. Riley and colleagues:

Thanks for revising your manuscript. The reviewers are mostly satisfied with your revision (as am I). Great! However, there are a few issues still to entertain. Please address these ASAP so we may move towards acceptance of your work.

Best,

-joe

Reviewer 2 ·

Basic reporting

This is much better than previous version. Authors made a really good job to improve the manuscript.

Experimental design

This is much better than previous version. Authors made a really good job to improve the manuscript.

Validity of the findings

This is much better than previous version. Authors made a really good job to improve the manuscript.

Additional comments

I am happy to see the improved version of this manuscript. Now it is really worth to be published almost as it is. There are only a few minor issues in the manuscript:

line 27: Results: "only 139 species in 47 genera" - when reader does not know how many species and genera are in Lampyridae, it is difficult to understand if 139/47 is a small or high number...

line 73: Methodology: "lampirid" really lampirid or lampyrid?

line 165: bioluminescence in Elateroidea - authors cite only Bocakova et al. (2007), however, to cover also more lineages and new findings I would suggest citing also e.g., Bi et al. (2019) ZooKeys and Fallon et al. (2018) eLife.

line 275 - "emit bioluminescence" better: emit light?

line 283 - Linnaeus, 1758 should not be italicized

line 293 - "larval bioluminescent" should be probably "larval bioluminescene".

line 376 - "))" instead of ")" after Rosa et al. 2020

line 444 - e.g. - a dot is missing

line 454 - lampyrid fireflies sounds odd to me...just fireflies would be enough

line 455 - lampyroid should always be within quotation marks as this is not the real superfamily but only an informal name for the clade consisting of fireflies and relatives

Reviewer 3 ·

Basic reporting

No comment

Experimental design

No comment

Validity of the findings

No comment

Additional comments

The manuscript has improved substantially and most of the reviewers' questions and concerns have been addressed. However, there are a few points the authors may need to double-check before the manuscript can be published:

73 Please check if this is a typo. "Lampiri" or "Lampyri"?

155 heavily hardened sclerites; no, not in the aquatics

156 they describe shape as if it is a box with top bottom and sides but that is not the shape it is more like a top and bottom, the bottom including the lateral elements

217 NOT pleurae. Singular is pleuron, pleural is pleura

222 NOT Pyrophanes. A Pyrophanes larva having gills is totally unsubstantiated

224 Use currently accepted name. Sclerotia substriata (Gorham, 1880)

228 not Aquatica it is Sclerotia

336 Aquatica larvae can move, not sure if they swim, across the bottom of ponds

337 Sclerotia larvae actually swim upside down just below the water surface and parallel to it with just the tip of the abdomen and thus the terminal spiracles piercing the water surface. They are back swimmers as they swim on their backs but they still swim towards their head end

360 Typo: "void". Change it "to avoid"

369 Check Fu & Ballantyne 2008 where they noted Pygoluciola qingyu will attack ants head on

Figure 1 Genera names should be italicised
Figure 2 Same comment as the above. Scientific names should be italicised

---

## Round 0.3 · accepted · Accept

Dear Dr. Riley and colleagues:

Thanks for revising your manuscript based on the concerns raised by the reviewer. I now believe that your manuscript is suitable for publication. Congratulations! I look forward to seeing this work in print, and I anticipate it being an important resource for groups studying firefly systematics. Thanks again for choosing PeerJ to publish such important work.

Best,

-joe